# The Influence of the COVID-19 Pandemic on Women’s Feelings during a Hospital Stay

**DOI:** 10.3390/ijerph19116379

**Published:** 2022-05-24

**Authors:** Agnieszka Kułak-Bejda, Grzegorz Bejda, Elżbieta Krajewska-Kułak, Anna Ślifirczyk, Joanna Chilińska, Alicja Moczydłowska, Napoleon Waszkiewicz, Marek Sobolewski

**Affiliations:** 1Department of Psychiatry, Medical University of Białystok, 16-070 Choroszcz, Poland; napoleonwas@yahoo.com; 2Faculty of General Medicine, School of Medical Science in Bialystok, 15-875 Bialystok, Poland; grzegorzbej@gmail.com; 3Department of Integrated Medical Care, Faculty of Health Sciences, Medical University of Białystok, 15-096 Białystok, Poland; elzbieta.krajewska-kulak@umb.edu.pl; 4Department of Emergency Medicine, Faculty of Health Sciences, Pope John Paul II State School of Higher Education in Biała Podlaska, 21-500 Biała Podlaska, Poland; aslifirczyk1@gmail.com; 5Faculty of Health Science, Łomża State University of Applied Sciences, 18-400 Łomża, Poland; joasia.chilinska@op.pl; 6Academy of Agrobusiness in Lomza, The Siedlce University of Natural Sciences and Humanities, 08-110 Siedlce, Poland; amoczydlowska.wsa@wp.pl; 7Department of Quantitative Methods, Rzeszów University of Technology, 35-959 Rzeszów, Poland; msobolew@prz.edu.pl

**Keywords:** COVID-19, stress, hope, general self-efficacy, depression, women

## Abstract

**Introduction:** The COVID-19 (coronavirus disease 2019) affected individuals and society and caused disruption, anxiety, stress, and loneliness. Being hospitalized during the pandemic increase a patient’s negative feelings. This study aimed to evaluate the impact of the COVID-19 pandemic on patients’ feelings (loneliness, depression, hope, self-efficacy) during a hospital stay. **Materials and methods:** This study included 207 women, aged from 15 to 83 years (55 ± 21.2) that were hospitalized during the pandemic in Białystok, Łomża, and Biała Podlaska, Poland at internal medicine departments. The main reasons for hospitalization were cardiovascular diseases, abdominal pain, chronic obstructive pulmonary disease, pneumonia, diabetes, and unknown fever. Respondents were asked to complete the following questionnaires: Sense of Loneliness (the DJGLS), Depression Beck Inventory (BDI), Basic Hope (BHI-12), and Self-Efficacy (GSES). **Results:** Most of the studied group of women had an average sense of loneliness. A greater sense of loneliness was found among hospitalized women living in the countryside—the lowest sense of loneliness and depression was among women with higher education and the highest sense of effectiveness. One-third of respondents had a moderate degree of depression. Of the respondents, 39% had a relatively high level of basic hope. The assessment of self-efficacy demonstrated that 52% of the respondents showed a high sense of self-efficacy, an average sense of self-efficacy was shown by 35.5% of the respondents, and a low sense of self-efficacy was shown by 12.6% of the respondents. **Conclusions:** Numerous hospitalized women during the COVID-19 pandemic, despite experiencing moderate depression, had an average sense of loneliness and a high level of hope and self-efficacy.

## 1. Introduction

Coming to a hospital and staying in it is a difficult experience for most patients and is associated with, among other things, uncertainty about the essence of their disease, detachment from their usual lifestyle, the strangeness and their ignorance of the environment, a sense of losing control throughout events, and a loss of privacy. During hospitalization, there is a loss of independence and a dependence on unknown people and pain caused by disease. In addition to disease symptoms, the patient can develop a sense of danger and a loss of sense of security. A source of discomfort for a patient staying in a hospital is the loss of privacy due to examination and medical procedures. The patient enters a new environment, becomes one of many, and feels anxious. As a consequence, the above can make the person depressed and they may even become depressed [1]. The mental state is an element of the mental process, which includes the following, among others: emotional state (affect and mood), psychomotor drive, cognitive processes, self-perception, and behavior. It includes both elements of wellbeing (optimal psychological functioning and positive experiences) and the absence of pathological states (e.g., anxiety or depression) and pathological symptoms [2,3].

Disease and its effects are already stressors, and the necessity to stay in the hospital increases the current mental burden [4,5].

Rosselli et al. [6] found that higher anxiety and depression levels were associated with urogenital, rheumatologic, oncological, neurological, and respiratory disorders. Higher distress scores were associated with rheumatologic and digestive system illnesses, while a higher severity of distress was associated with oncological illnesses.

It is worth remembering that stress can be mobilizing or destructive. However, both forms do not differ physiologically because they activate the human body with the same mechanisms. However, directly related to destructive stress is stress, which is associated with a number of emotional and psychological reactions, ranging from anger and anxiety to depression [6].

The above was significantly exacerbated during the COVID-19 pandemic. During the COVID-19 pandemic, general medical complications have received the most attention [7]. Furthermore, the indirect effects of the pandemic on general mental health were of increasing concern. Studies of the general public revealed lower psychological wellbeing and higher scores of anxiety and depression compared to before the COVID-19 pandemic.

Humans have many mechanisms for coping with stressful situations and depression; hope is one of them. Hope correlates positively with wellbeing and negatively with anxiety, depression, and psychosomatic symptoms [8].

The feeling of hope is significantly inscribed in the life of every human being. Motivating and dynamizing a person’s actions contributes to their comprehensive and full development. On the other hand, lack of hope often leads to apathy, boredom, frustration, and despair. Thus, in a person’s everyday life, they “create” the world as a more human reality. The emphasis on having hope means the constant internal readiness of the individual to face the challenges that are to come. In this sense, hope is an expression of intense—though still unfulfilled—human activity. Being associated with the emotional and cognitive sphere accompanies humans at all stages of their development [8].

According to Erikson, hope is “an expectant desire”, a driving force that arises from resolving the first developmental conflict between trust and distrust in the first year of life [9]. Continuing Erikson’s thought, Trzebiński and Zięba proposed the use of the term “basic hope” to describe hope. Basic hope is considered a fundamental constituent of an individual’s worldview, which is mostly unconscious and learned very early. It also influences how the individual reacts to new, unknown situations. For example, people with a high level of basic hope, more often than people with a low level of hope, perceive difficult life situations as a challenge and an opportunity for development. Basic hope seems to predict positive effects in psychotherapy; it correlates positively with wellbeing and negatively with anxiety, depression, and psychosomatic symptoms. The Basic Hope Inventory (BHI) was developed to measure the strength of basic hope [10]. Hope is often activated during illness, which is a medical problem and a psychosocial one. It is a new situation that an individual has to deal with in a certain way. Then, a person undertakes an activity defined as coping, aimed at the self-control of emotions and controlling the source of stress [11].

Women report more loneliness than men, irrespective of age. Although, women are socialized to develop a larger and more active social network, potentially protecting them from loneliness. Moreover, women live longer than men and are more likely to be affected by widowhood or assume the role of caregiver for their spouse [12]. The prevalence of depression is higher in women than in men. For example, in 2010 the global annual prevalence of depression had a 1.7-fold greater incidence in women [13]. This sex disparity indicates a potential role for gonadal hormones in the etiology of anxiety and depressive disorders. In addition, studies have revealed that women are more likely to experience mood disturbances, anxiety, and depression during hormonal flux, such as puberty, menopause, and perimenstrual and post-partum periods [14]. Hormonal flux in females appears to increase the likelihood of experiencing mood disturbances; clinical and preclinical studies in males suggest that testosterone has protective benefits against anxiety and depression. It is suggested that testosterone mediates such protective effects, including the brain sites, biochemical factors, and molecular pathways [15].

Bandura has defined self-efficacy as people’s beliefs in their capabilities to exercise control over their functioning and events that affect their lives [15]. One’s sense of self-efficacy can provide the foundation for motivation, wellbeing, and personal accomplishment. High self-efficacy has been linked with numerous benefits to daily life, such as resilience to adversity and stress, healthy lifestyle habits, improved performance at work, and educational achievement.

In Poland, the first case of coronavirus infection was detected on 4 March 2020. Since then, SARS-CoV-2 has appeared in all provinces. Due to the pathogen’s spread in the country, an epidemiological emergency was declared on 13 March 2020. However, on 20 March 2020, it was canceled, and an epidemic was introduced, which has continued up until the time of writing this work.

Hospitals were subject to a strict sanitary regime during this time and visits were prohibited. Therefore, the contact of the hospitalized patient with their relatives was limited only to a telephone conversation.

The current study continued our interest in women’s mental health during pregnancy, after childbirth, and menopause. Unfortunately, to our best knowledge, no previous studies have been performed on the impact the COVID-19 pandemic on adult female patients’ feelings (depression, hope) during a hospital stay.

## 2. The Objective of the Study

The study aimed to assess hospitalized women’s feelings (loneliness, depression, hope, self-efficacy) during the pandemic. We also analyzed correlations between women’s depression, hope, self-efficacy, and demographic data (age, place of residence, and education). The research hypothesis assumed that women who felt more lonely than usual due to the pandemic situation had a lower level of essential hope and a lower sense of self-efficacy.

## 3. Materials and Methods

The study included 207 women that were hospitalized during the pandemic in hospitals in Białystok, Łomża, and Biała Podlaska, Poland at internal medicine departments. The main reasons for hospitalization were cardiovascular diseases (heart failure, atrial fibrillation, myocardial infarction, hypertension), abdominal pain, chronic obstructive pulmonary disease, pneumonia, diabetes, and unknown fever. Participants were not infected with COVID-19.

The research was conducted after obtaining the approval of the Bioethics Committee of the Medical University of Bialystok, Poland, APK.002.304.2020 and hospital managements in September and October 2020.

The inclusion criteria included (1) women over the age of 18 years old, (2) women without depression in their history, (3) women not treated for psychiatric disorders, (4) women who were able to fill out the questionnaires. The exclusion criteria included (1) women aged less than 18 years old, (2) women with depression, (3) women treated for psychiatric disorders, (4) women who were not able to fill out the questionnaires.

A total of 220 questionnaires were distributed, and 207 fully completed questionnaires qualified for the study. Thirteen questionnaires were not fully completed. The response rate was 94%. The analysis included 207 women hospitalized during the “pandemic”, and its purpose was to determine how they assessed their hospital stay at that time. The pandemic forced many women to take a care allowance for children up to eight years of age or, simply, a holiday. Additionally, the stay in the hospital contributed to the aggravation of the problems mentioned above.

The study used four standardized scales.

The standardized scale for measuring the sense of loneliness by De J. Bending (the DJGLS) description, reporting on relationships with 11 items including final reports, ad hoc statements, reports, and the remaining five—finally financed—items measured satisfaction with interpersonal relationships. The respondent was asked to indicate the statement in the description that expressed her current allocation and feelings. Responses were given on a five-point scale from “definitely yes” to “definitely no”. The loneliness index was calculated by recoding six “negative” items (i.e., statements 2, 3, 5, 6, 9, and 10) and then summing up all test items. The maximum number of points was 55, and the higher the respondent’s total score, the greater their sense of loneliness. Cronbach’s alpha internal stability coefficient of the scale was high (α = 0.89), as was the value of the mean inter-position correlation (r = 0.42) and the Loevinger H homogeneity coefficient (H = 0.47). Cronbach’s alpha internal stability coefficients were high and amounted to 0.89 [16].

The Standardized Basic Hope Questionnaire (BHI 12) [10] consists of 12 statements, 9 of which are diagnostic, and the remaining 3 (1, 4, 7) are buffer statements. The total possible score ranges from 9 to 45 points. The higher the score, the greater the hope. Diagnostic statements refer to beliefs about the benevolence of the world and the order and predictability of the world. The test person’s task is to assess how they agree with particular statements on a five-point scale. 

The Beck Depression Scale consists of 21 points assessed according to the intensity of symptoms, from 0 to 3. From each point, the subject chooses one answer which, in their opinion, best describes their condition in the indicated period. For example, scoring 0–10 points shows no depression, 11–16 points—mild depression, 17–20—mild mood disorders, 21–30—borderline clinical depression, 31–40—moderate depression, over 40—severe depression [17].

The General Self-Efficacy Scale (GSES) is correlated to emotion, optimism, work satisfaction. Negative coefficients were found for depression, stress, health complaints, burnout, and anxiety. The total score is calculated by finding the sum of the all items. For the GSE, the total score ranges from 10 to 40, with a higher score indicating more self-efficacy. Internal reliability for GSE is equivalent to a Cronbach’s alpha between 0.76 and 0.90 [18].

## 4. Statistical Analysis

Statistical analysis was performed. First, Spearman’s rank correlation was used to test the relationship between two numerical features. The values of the mean and the median values for psychometric measures in relation to the place of residence and then the level of education were compared, and the significance of differences (at *p* < 0.05) between the groups were assessed using non-parametric tests, the Mann-Whitney test or the Kruskal–Wallis test because certain measures deviated from normality. The confidence interval was 95%. All statistical analyses were performed with Statistica 13 P.L.

## 5. Results

In 62.8% of patients, on admission to the hospital, the COVID-19 test was performed, and in the remaining 37.2%, the test was not performed because they had to do it earlier. All of them had negative results.

The age of the surveyed patients ranged from 15 to 83 years old (55 ± 21.2 years). The age distribution was also divided into ten-year age ranges. The studied group was dominated by patients aged 60–69 (29%), 50–59 years old—19%, 30–39 years old—15%, 70–79 years old—14%, 40–49 years old—11%, 20–29 years old—8%, over 80 years old—3%, and under 20 years old—1%. The vast majority of the surveyed patients were married (64.7%). This was followed by widows (15.9%), unmarried women (9.7%), people who were divorced (5.3%), and people who were in an informal relationship (3.3%) or who were separated (1%).

Approximately half of the surveyed patients (51.7%) were residents of cities and 48.3% of surveyed patients were residents of rural areas.

Patients’ opinions about their stay in the hospital were collected as answers to two groups of questions (positive and negative meanings) from the author’s part of the questionnaire. First, respondents rated the intensity of experiences related to their stay in the hospital on a scale from 0 to 5 points.

Both groups of questions were analyzed separately to facilitate the ranking of the results. In addition, several of the most negative aspects of a hospital stay during the pandemic (which were most lacking during a hospital stay) were asked to be identified.

The main problems were the lack of visitors (59.4%) and the inability to move freely (43.5%). In addition, approximately every third patient developed fear (29.5%), which resulted from the atmosphere of danger, fear of the virus, and the lack of reliable information on infections (10.1%). Other problems (poor food, lack of proper care) were indicated by 14.5% of the respondents, and 4.8% said they had no problems.

In the “ranking” of negative feelings (on a scale from 0 to 5) during the hospital stay, the severity of the limitation of family visits and the hospitalization during the pandemic dominated. In addition, among the positive ones, it was noted that there was support from the family despite the lack of contact and emotional support from nurses (Table 1).

The mental condition of the patients was assessed using four standardized questionnaires: Sense of Loneliness (de Jong Gierveld—the DJGLS), Depression Beck Inventory (BDI), Basic Hope (BHI-12), and Self-Efficacy (GSES).

Total psychometric measures were determined based on the answers given to the component questions included in each of these questionnaires—the classification of results based on the standards proposed by the authors of the scales. The studied group of women showed the following (Table 2):level of loneliness—average, within 26.7 points (out of a possible 55), which proved its average level;depression level—average, 9.7 points (out of a possible 63), the lack of it concerned 64.2% of the respondents but, in the group of 30.7%, the degree of depression was still moderate and in 5% it was severe;level of basic hope—average, 39.5 points (out of a possible 45), which proved a relatively high level of basic hope;self-efficacy—29 points on average, 6 sten (out of 40 possible points, 10 sten), and 51.9% of the respondents showed a high sense of self-efficacy, an average sense of self-efficacy was shown by 35.5% of the respondents and low sense of self-efficacy was shown by 12.6% of the respondents.

Statistically significant, but not very strong correlations, were shown between the measures (the test probability *p* values were below 0.05 for all dependencies), as shown in Table 3.

It was also examined whether factors such as age, place of residence, and education differentiated the mental condition of patients. As the distribution of measures differed from the normal distribution (e.g., for BDI), which was confirmed by the results of the Shapiro¬–Wilk test (apart from the BHI-12 measure where statistically significant deviations from the normal distribution were found), non-parametric statistical tests were used for the analysis at this point.

The Spearman correlation coefficients between age and the considered psychometric measures were determined. There was a statistically significant correlation between age and BDI and GSES and between age and the measure of loneliness. With age, the level of depression and loneliness increased, and the sense of self-efficacy decreased. However, these were very weak correlations (Table 4).

Selected correlations are presented in the scatter plots (Figure 1).

In the case of the residence, a statistically significant difference was found only in the sense of loneliness, which was more significant among hospitalized people living in the countryside (on average, 28.3 vs. 25.3; *p* = 0.0315)—Table 5.

The mental condition of patients was more diversified in terms of education. Women with higher education were characterized by the lowest sense of loneliness and depression and the highest sense of effectiveness. However, for the measure of loneliness, the difference was highly statistically significant, and for the other two measures it was only close to the level of significance. In addition, due to the small group of people with undergraduate education, they were included in the group of people with higher education (Table 6).

The distribution of the measure of loneliness concerning the level of education is presented in the following diagram (Figure 2).

The level of psychometric measures was also compared to the department where the patients were hospitalized. The reason for carrying out this analysis was the assumption that contracting COVID-19 may harm the psyche of patients. No significant relationships were found between the two groups in terms of the values of psychometric scales (Table 7).

The mental condition of the patients was compared depending on the fact that the test for COVID-19 was performed. A slightly stronger sense of loneliness (difference at the borderline of statistical significance: *p* = 0.0689) was demonstrated by patients who underwent the COVID-19 test immediately upon admission to the hospital (on average 27.5 vs. 25.4 points). The results are presented in Table 8.

An analysis was also carried out to divide persons with a negative test result and a positive result. It was interesting to check whether there would be any differences in such a division—whether people with a negative test result would show a better mental state because, theoretically speaking, they should get rid of the fears related to the epidemic.

On the other hand, taking the test may raise a negative mental attitude towards hospitalization. As can be seen, there was clearer, similar to statistically significant differences between the compared groups, only for the measure of loneliness. The opposing group of people with negative test results was different. If we only compared this group to the combined two, we would have obtained a *p*-value of 0.0176, which was a statistically significant difference. The results are presented in Table 9 and Figure 3.

## 6. Discussion

In the present study, we assessed the feelings of women hospitalized during the pandemic. The main problems were the lack of visitors and the inability to move freely. In addition, patients developed a fear of the virus and lack of reliable information on infections.

However, in the current research, among respondents, loneliness was at the average level.

In contrast, in a nationwide study of a sample of American adults from 2020, there were no significant mean-level changes in loneliness across the three assessments (January/February 2020 to April 2020). However, older adults reported less loneliness compared to younger adults [19].

A similar study from the U.S., which included 1013 adults, completed the UCLA Loneliness Scale-3 and Public Health Questionnaire. In 43% of respondents, loneliness was elevated and was strongly associated with more significant depression and suicidal ideation [20].

Golaszewski et al. [21] examined the associations of social isolation and loneliness with incidents of cardiovascular disease (CVD) in a large cohort of postmenopausal women and whether social support moderated these associations. The study was conducted from March 2011 through March 2019 and included community-living U.S. women. In this study, social isolation and loneliness were independently associated with modestly higher CVD risk among postmenopausal women in the U.S. Women with social isolation and loneliness had greater CVD risk than those with either exposure alone.

A Norwegian study conducted in 2020 explored trends in loneliness before and during the COVID-19 pandemic [22]. Overall, loneliness was stable or falling during the lockdown. However, single individuals and older women, reported slightly increased loneliness during the lockdown. These findings were consistent with our results.

The rate of depression among respondents varied in many reports from Europe, Asia, and America. In the present study, one-third of respondents had a moderate degree of depression. Therefore, our findings were in accordance with previous reports. In a Polish study conducted four times (in May, June, July, and December 2020) as part of a nationwide project on the psychological aspects of the COVID-19 epidemic [23], symptoms of depression, and anxiety were assessed. In December 2020, 29% of women and 24% of men had symptoms of depression. In the current study, conducted in September and October 2020, 30.7% of the women surveyed had moderate depression and 5% severe depression. Our results were in accordance with previous studies.

In a cross-sectional study conducted in Germany of 15,704 German residents aged 18 years and over, 14% had depression. Moreover, females and younger people reported more mental health problems [24]. In a Chinese study of 1109 individuals, 12% of cases had depression, and 19% had anxiety. Logistic regression analysis demonstrated that gender, age, education of parents, com weekdays, and physical exercise were significantly associated with depression and gender, physical exercise, and companionship on weekdays were significantly associated with anxiety [25]. In a Hong Kong population-based study, in a sample of 500 respondents, 19% had depression, and 14% had anxiety. Furthermore, 25% reported that their mental health had deteriorated since the pandemic [26]. In a cross-sectional online study that assessed 898 young adults (18–30 years old) from the U.S., 43% respondents reported high levels of depression, 45% had high anxiety scores, and 32% had high levels of PTSD symptoms. Furthermore, high levels of loneliness, high levels of COVID-19-specific worry, and low distress tolerance were significantly associated with clinical levels of depression [27].

In the current study, 39.5% of respondents had a relatively high level of hope. In a similar report from Israel that compared hope levels among a sample of 584 adults (2020 survey) to 884 Internet users, which was performed six months prior to the COVID-19 pandemic lockdown, age, gender, and education did not affect these differences. Furthermore, high degrees of depression and hope levels significantly increased in the 2020 survey compared to the 2019 survey [28]. Basic research in personality, social, and developmental psychology established the causal impact of self-efficacy and goal processes on social behavior and wellbeing [29]. The assessment of self-efficacy in the present research demonstrated that 52% of the respondents showed a high sense of self-efficacy, 35.5% of the respondents showed an average level of self efficacy, and 12.6% of the respondents showed a low level of self-efficacy. Ritchie et al. demonstrated a significant drop in self-efficacy beliefs from before to during the early period of the lockdown in the COVID-19 crisis [30].

Marcysiak et al. [31], in a group of 108 patients from the Provincial Hospital in Ciechanów, Poland, demonstrated that the level of hope among hospitalized patients was significantly lower than in the control group. Furthermore, men had a higher level of hope. In contrast, in the present study, the level of hope of the surveyed women was reasonably high.

When faced with an illness, a person activates various behaviors aimed at coping with it. These strategies are primarily determined by the cognitive attitude towards one’s illness and the way of experiencing it emotionally. They include two interpenetrating and interacting forms: self-regulation of emotions, reducing negative emotions and stimulating positive feelings, and task-oriented remedial activity to improve health. In the area of these behaviors, various and individualized styles of action are observed [32]. Therefore, in illness and hospitalization, a patient’s effectiveness level seems to be important. It is believed that a high sense of self-efficacy is the belief that there is a possibility of coping with a given situation, even in unfavorable circumstances [33]. It is suggested that self-efficacy is associated with better wellbeing; it determines the taking of action, the effort put in it, the feelings accompanying it, perseverance in pursuing the goal, coping with obstacles and failures, and experiencing stress. In addition, a high level of self-efficacy increases our resources of stress resistance. Krok et al. [34] demonstrated that the relationships between the risk of contracting COVID-19, personal resources, and subjective wellbeing had an indirect impact and could include mediating factors related to meaning-making processes and stress experiences. The central finding demonstrated the different mediating roles of stress and meaning making in the relationship of risk of contracting COVID-19 and personal resources with the cognitive and affective dimensions of subjective wellbeing. At the same time, based on feedback, successes increased the sense of self-efficacy, which brought beneficial effects not only in general coping with stress but also with the disease [35,36,37,38,39]. In the present study, the surveyed women showed an average level of self-efficacy on the scale with 29 points out of a possible 40. Furthermore, 51.9% of the respondents, showed a high sense of self-efficacy, 35.5% of the respondents showed an average level of self-efficacy, and 12.6% of the respondents showed a low level of self-efficacy. It was also found that self-efficacy decreased with age. Women with higher education showed the highest sense of effectiveness.

People whose COVID-19 infection was confirmed by the test did not differ in the severity of symptoms of depression and anxiety from the other two groups [22]. In the present study, no significant relationships were found regarding the values of the psychometric scales between the groups of patients from non-COVID-19 and COVID-19 wards, but this might be due to the small number of patients hospitalized in COVID-19 wards. Only for measure of loneliness was clearer, similar to statistically significant differences between the compared groups. The group of people with negative test results was out of the question. However, if we only compared this group to the combined two, we would get a *p*-value of 0.0176, a statistically significant difference. Perhaps people with a negative test result had a stronger feeling of loneliness because they had to stay home. And people with positive the COVID-19 infection had more often stay in a hospital.

## 7. The Study Limitations

Some limitations to our study need to be acknowledged. First, a relatively small sample size was evaluated. Second, the study group included only women, and we did not analyze men. It is worth emphasizing that women are nearly twice as likely as men to be diagnosed with depression. Third, our results were limited by the use of a cross-sectional study design. Fourth, respondents were recruited from different departments. We did not used the Hospital Anxiety and Depression Scale in the present study. This scale was more appropriate for assessing anxiety and depression in a hospital setting. Patients were hospitalized for different disorders: cardiovascular diseases, gastrointestinal and pulmonary disorders, diabetes, and unknown fever. It was a potentially confounding variable, and might be a serious limitation of the study to have mixed participants. Finally, the impacts of COVID-19 lockdowns may be different among countries; thus, generalization of the results may be difficult.

## 8. Conclusions

The dominant negative feelings related to a hospital stay during the pandemic among the surveyed women were the limitation on family visits and the need for hospitalization.

Most of the studied group of women had an average level of a sense of loneliness, a relatively high level of essential hope and sense of self-efficacy, and one-third of the respondents showed a moderate or severe degree of depression.

Among hospitalized women living in the countryside, a greater sense of loneliness was found, and among women with higher education the lowest sense of loneliness and depression was recorded along with the highest sense of effectiveness

Patients demonstrated a stronger sense of loneliness with a negative COVID-19 test.

## Figures and Tables

**Figure 1 ijerph-19-06379-f001:**
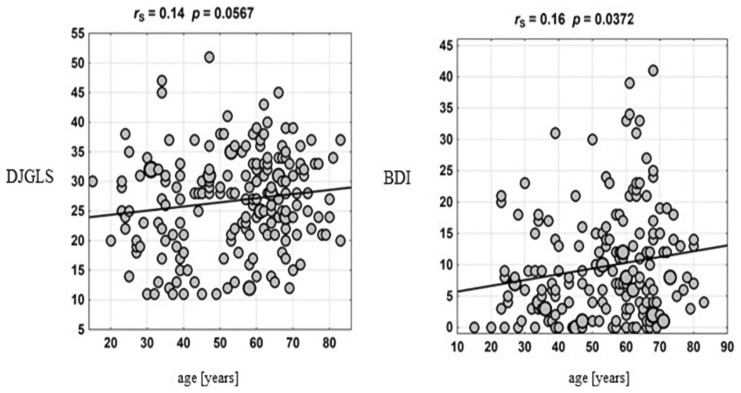
Correlations between the applied psychometric scales and age.

**Figure 2 ijerph-19-06379-f002:**
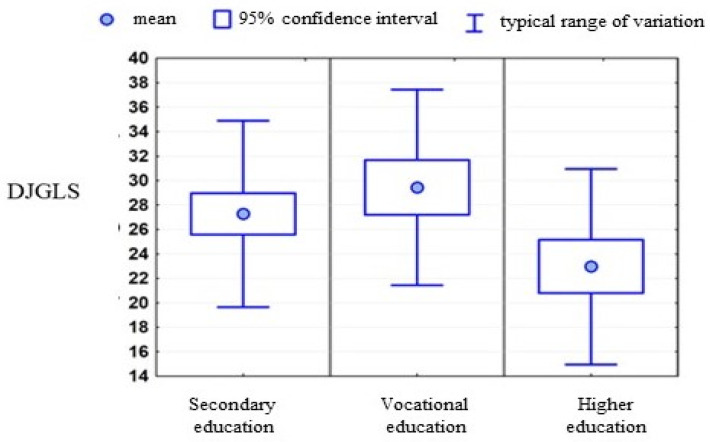
Correlations between the psychometric scales used and the level of education. Mean value +/− standard deviation.

**Figure 3 ijerph-19-06379-f003:**
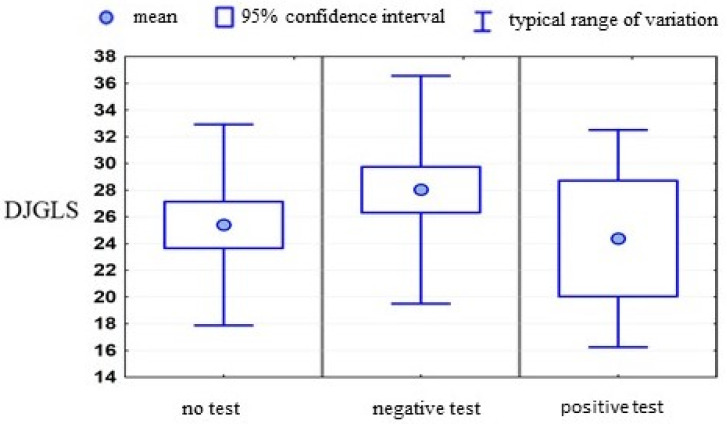
Correlations between the applied psychometric scales and the result of the COVID-19 test. Mean value +/− standard devaition.

**Table 1 ijerph-19-06379-t001:** Negative and positive aspects of a hospital stay, as assessed by the respondents.

	M	Median	SD
Negative aspects of staying in hospital			
The severity of limiting family visits during pandemic	3.34	4	1.81
The need for hospitalization during a pandemic	2.91	3	1.82
Conducting conversations on substitute topics	2.72	3	1.84
Running away from thoughts of the pandemic	2.60	3	1.83
Fear of getting COVID-19	2.56	2	1.82
A strategy to survive a hospital stay	2.24	3	1.77
How to avoid a COVID-19 infection	1.82	2	1.74
Care in the hospital is worse than before the pandemic	1.29	0	1.77
The feeling of giving up or surrendering	1.25	0	1.66
Willingness to escape from the hospital	0.81	0	1.62
Positive aspects of staying in hospital			
Receiving emotional support from the family despite the lack of direct contact	4.11	5	1.45
Receiving emotional support from nurses	3.75	4	1.53
Receiving emotional support from doctors	3.68	4	1.50
Care in hospitals is better than before the pandemic	3.19	3	1.71
Searching for the good sides of staying in hospital	2.66	3	1.78
Receiving emotional support from other patients	2.55	3	1.87

**Table 2 ijerph-19-06379-t002:** The results of the psychometric scales of the respondents.

Psychometric Features	M (with 95% c.i.)	SD	Median	Min	Max
DJGLS	26.7 (25.5–27.9)	8.2	27.5	11	51
Depression Scale Beck	9.7 (8.4–11.0)	8.6	7	0	41
BHI-12	39.5 (38.7–40.3)	5.7	39.5	24	59
GSES	29.0 (28.4–29.5)	3.8	30	17	40

M—mean; SD—standard deviation.

**Table 3 ijerph-19-06379-t003:** Correlations between the psychometric scales used.

Psychometric Features	DJGLS	BDI	BHI-12	PWS
**DJGLS**	1	0.26 (*p* = 0.0006)	−0.38 (*p* = 0.0000)	−0.23 (*p* = 0.0021)
**BDI**	0.26 (*p* = 0.0006)	1	−0.28 (*p* = 0.0003)	−0.42 (*p* = 0.0000)
**BHI-12**	−0.38 (*p* = 0.0000)	−0.28 (*p* = 0.0003)	1	0.23 (*p* = 0.0018)
**GSES**	−0.23 (*p* = 0.0021)	−0.42 (*p* = 0.0000)	0.23 (*p* = 0.0018)	1

**Table 4 ijerph-19-06379-t004:** Correlations between the applied psychometric scales and age.

Psychometric Features	Age [Years]
**DJGLS**	0.14 (*p* = 0.0567)
**BDI**	0.16 (*p* = 0.0372)
**BHI-12**	0.06 (*p* = 0.4099)
**GSES**	−0.18 (*p* = 0.0125)

**Table 5 ijerph-19-06379-t005:** Correlations between the applied psychometric scales and the place of residence.

Psychometric Features	Place of Residence				*p*-Value
	Village		City		
	M (95% c.i.)	Median	M (95% c.i.)	Median	
DJGLS	28.3 (26.5–30.1)	28	25.3 (23.8–26.8)	25	0.0315
BDI	9.3 (7.7–10.9)	8	10.0 (8.1–12.0)	7	0.9355
BHI-12	38.8 (37.5–40.1)	39	40.1 (39.1–41.2)	40	0.1056
GSES	28.9 (28.1–29.7)	30	29.0 (28.3–29.8)	30	0.8090

M—mean; SD—standard deviation. *p*-test probability values were calculated using the Mann–Whitney test.

**Table 6 ijerph-19-06379-t006:** Correlations between psychometric scales and the place of stay.

Psychometric Features	Education						*p*-Value
	Secondary		Professional		Higher		
	M (95% c.i.)	Median	M (95% c.i.)	Median	M (95% c.i.)	Median	
DJGS	27.3 (25.6–29.0)	27	29.4 (27.2–31.7)	30.5	23.0 (20.8–25.1)	23.5	0.0004
BDI	10.5 (8.5–12.6)	8	10.6 (8.1–13.2)	9	6.9 (5.0–8.8)	6	0.0515
BHI-12	38.5 (37.3–39.7)	39	40.3 (38.7–41.9)	40	40.3 (38.6–42.1)	40.5	0.1504
GSES	28.9 (28.1–29.8)	30	28.2 (27.0–29.4)	29	29.8 (28.8–30.7)	30	0.0748

M—mean; SD—standard deviation. *p*-test probability values were calculated using the Kruskal–Wallis test.

**Table 7 ijerph-19-06379-t007:** Correlations between the applied psychometric scales and the place of residence.

Psychometric Features	Hospital Department				*p*-Value
	Non COVID-19		COVID-19		
	M (95% c.i.)	Median	M (95% c.i.)	Median	
DJGS	26.9 (25.7–28.1)	28	24.4 (20.0–28.7)	26	0.2903
BDI	9.7 (8.4–11.1)	7	9.4 (5.8–12.9)	9	0.7695
BHI-12	39.3 (38.5–40.2)	39	41.4 (38.6–44.2)	42.5	0.1404
GSES	28.8 (28.3–29.4)	30	30.8 (28.2–33.4)	30	0.1610

M—mean; SD—standard deviation. *p*-test probability values were calculated using the Mann–Whitney test.

**Table 8 ijerph-19-06379-t008:** Correlations between the psychometric scales used and the place where the COVID-19 test was performed.

Psychometric Features	Testing for COVID-19				*p*-Value
	Immediately upon Admission to the Hospital		In Advance		
	M (95% c.i.)	Median	M (95% c.i.)	Median	
DJGS	27.5 (25.9–29.1)	28	25.4 (23.7–27.1)	25	0.0689
BDI	9.7 (8.0–11.4)	7	9.7 (7.8–11.6)	8	0.7755
BHI-12	39.5 (38.5–40.6)	40	39.5 (38.1–40.9)	39	0.9290
GSES	29.4 (28.6–30.2)	30	28.4 (27.7–29.0)	29	0.1446

M—mean; SD—standard deviation. *p*-test probability values were calculated using the Mann–Whitney test.

**Table 9 ijerph-19-06379-t009:** Correlations between the psychometric scales used and the result of the COVID-19 test.

Psychometric Features	Test/Result						*p*-Value
	No Test		Negative Test		Positive Test		
	M (95% CI)	Median	M (95% CI)	Median	M (95% CI)	Median	
DJGS	25.4 (23.7–27.1)	25	28.0 (26.3–29.8)	29	24.4 (20.0–28.7)	26	0.0565
BDI	9.7 (7.8–11.6)	8	9.7 (7.8–11.7)	7	9.4 (5.8–12.9)	9	0.8906
BHI-12	39.5 (38.1–40.9)	39	39.2 (38.1–40.4)	39	41.4 (38.6–44.2)	42.5	0.2974
GSES	28.4 (27.7–29.0)	29	29.2 (28.3–30.0)	30	30.8 (28.2–33.4)	30	0.1845

M—mean; SD—standard deviation. *p*-test probability values were calculated using the Kruskal–Wallis test.

## Data Availability

Data of this study are available on the request.

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
