# Peer review of "The Influence of the COVID-19 Pandemic on Women’s Feelings during a Hospital Stay"

_ijerph, 2022, doi:10.3390/ijerph19116379_

Round 1

Reviewer 1 Report

File attached

Author Response

Dear Reviewer, the topic of our manuscript does not apply to 'Comorbidities in school-aged children with cerebral palsy in Nigeria”. The title of our study is: ‘Influence of COVID-19 pandemic on women' feelings during a hospital stay’. Your review does not apply to our study. Yours faithfully.

Reviewer 2 Report

For the authors:

This manuscript addresses an important problem, Influence of COVID-19 pandemic on women' feelings during a hospital stay. Although the study has the potential to contribute information to this research field, there are several problems with the manuscript that would detract from its usefulness to readers. However, the details of the study are limited and need additional clarification. The following are more detailed comments.

Abstract

  1. The abstract, particularly the methods are weak in that it doesn't focus in depth on the main points, key features of the study. This section did not sufficiently describe the paper. The results seem to be mostly known conditions such as “During hospitalization, the main 11 problems were the lack of visitors (59%) and the inability to move freely (43%)”.
  2. The first time the authors use an abbreviation in the text, present both the spelled-out version and the short form (e.g., COVID-19, BHI-12, GSES) here.
  3. Please use the correct titles for outcome measures, such as: BDI should be “Beck Depression Inventory, not “Depression Beck”. The abbreviation and inappropriate descriptions make this abstract difficult to read and harder to understand.
  4. Abstract is where the authors could grab the reader’s attention. However, it is not clear and hard to read. Please follow the rules for correct formatting of your abstract.

Introduction and background

  1. Here as well as elsewhere in introduction would be stronger if showed evidence of a broader review and understanding of what is a large literature about the research topic. The manuscript could benefit from a comprehensive and well-organized review of the literature. The background section as it now stands weak.
  2. Background information is insufficient to help readers to obtain a better understanding of the research problem being investigated. For example, there is even no any information of self-efficacy. A more concisely articulated problem statement that promises to address a gap clearly identified in the background here would improve the paper. Please explain the scientific background and rationale for the investigation being reported.
  3. The relationships and context of COVID-19 pandemic, women, anxiety, depression, hope and self-efficacy are insufficient and not clear. Every parts seem to be independent, not link well.
  4. Some information or statements may need updated. The references are old. This is not appropriate. For example, No. 1 reference is a 2008 study.
  5. Some descriptions in the introduction are actually difficult to understand. For example, Line 26 “…the patient develops a sense of danger”; Line 36 “The disease and its effects ….”; Line 38-40 “However, both forms do not ….. and anxiety to depression”, etc. This is a significant study weaknesses.
  6. The study purpose is awkwardly worded and even not consistent with the purpose in abstract (e.g., self-efficacy).
  7. Please explain why choose women under the influence of COVID-19 pandemic.

Methods

The methods section also has some problems that detract from the manuscript.

  1. Please provide the eligibility criteria (inclusion and exclusion criteria), and the sources and methods of selection of participants.
  2. Please describe the sampling process. The recruitment process is not presented and this may limit generalizing the findings. How were participants approached? The description of data collection is missing as it does not give the reader a good sense of how recruitment actually unfolded.
  3. The reliability and validity of the study should be defined on the criterions suitable for quantitative methodology.
  4. Line 119-125: These results should not appear here.
  5. How do the authors measure “anxiety”? In this study, if I am not wrong, there is no single and reliable outcome measure for this variable.

Results, Discussion

  1. What are “groups”?
  2. The most problematic is the presentation of the research findings. For example, Line 176-179 “Patients' opinions about their stay in the hospital were collected as answers to two groups of questions from the author's part of the questionnaire. First, respondents rated the intensity of experiences related to their stay in the hospital on a scale from 0 to 5 points.” ---- (1). These may not belong to results. (2). First …. Then, what are the “second”?
  3. The tables need re-organized as data presentation is fragmented.
  4. The discussion is not thoroughgoing enough. There was not sufficient in-depth analysis and discussion of the study findings, or clinical uses thereof. The clinical value appears limited. For example, there is a remarkably high rate of depression. The mechanisms or explanations or discussion need to be sufficiently provided. Not just compare all the percentage of other studies with the present study. For example, whether some of participants are already have depression, regardless of COVID-19. This will be linked to the inclusion and exclusion criteria.
  5. The discussion section contains some useful information and provides some additional context for the findings, but it needs more information and better organization so that the reader is able to ascertain how the reported findings contribute to, extend, or challenge existing findings. The discussion section as it now stands weak and provides limited information.
  6. The authors did not sufficiently report the limitations of their study. For example, “only women” and this will bring up what limitations or presentation of the study findings.

Author Response

Thank you for your report. Please find the response attached

Reviewer 3 Report

Introduction,

Line 21. great experience- great should be replaced with another word.

Line 93-do not use personal language. There were also previous studies.

Methods

Who is the target group? Both covid and non-covid patients?

What is the sampling frame? How was the sample size calculated? Was it based 95% CI and 5% error?

Seventh para- line 131, 11 should be in bracket?

Line 164-166- There is no need to mention who did the analysis.

Line 166- Spearman rank/rho was used to test relationship between two numerical or two continuous variables? Please correct

Line 169- grammar, not groups was, groups were

All tests described were non-parametric

Results

Very confusing

Line 180- which two groups? COVID and non-covid?

Only the mean score values were given for depression and other measures?

What were the cut-off values to decide whether patients has anxiety or depression?

Discussion

Line 309 discusses metal health of emergency workers which is not relevant to the target group studied in this study.

Line 313 onwards- Were these studies for COVID-19 patients? The results are not comparable then.

Author Response

Thank you for your valuable comments

Introduction,

Line 21. great experience- great should be replaced with another word.

We have changed this word (Introduction)

Line 93-do not use personal language. There were also previous studies.

We have changed this (Introduction)

Methods

Who is the target group? Both covid and non-covid patients?

Participants were not infected with COVID-19 (Material and Methods, Results)

What is the sampling frame? How was the sample size calculated? Was it based 95% CI and 5% error?

CI was 95%, and 5% error. (statistical analysis)

Seventh para- line 131, 11 should be in bracket?

We have corrected this.

Line 164-166- There is no need to mention who did the analysis. –

We have removed this part.

Line 166- Spearman rank/rho was used to test relationship between two numerical or two continuous variables? Please correct

Yes, Spearman rank test was used.

Line 169- grammar, not groups was, groups were

We have corrected this mistake.

All tests described were non-parametric

They were described due to non normality distribution data.

Results

Very confusing

Line 180- which two groups? COVID and non-covid?

No, it was about dividing the questions into two groups (positive and negative meaning). (Results)

Only the mean score values were given for depression and other measures?

Not only the mean score values were given for depression , also for  other scales mean scores were provided (see Table 2).   

What were the cut-off values to decide whether patients has anxiety or depression?

The values were for depression.

Discussion

Line 309 discusses metal health of emergency workers which is not relevant to the target group studied in this study.

We have removed this article in the disccusion

Line 313 onwards- Were these studies for COVID-19 patients? The results are not comparable then.

These studies assessed eg. COVID-19-related fear, anxiety, depression and psychological distress and were conducted in population but not COVID-19 patient.

Reviewer 4 Report

The article explores a topical topic: assessing the feelings of women who were hospitalized during the pandemic with anxiety, depression, hopes for recovery and future self-efficacy. The authors proposed a study hypothesis that suggested that women who feel lonelier than usual due to the pandemic situation have a lower level of substantial hope and a lower sense of self-efficacy. The authors analyzed correlations between a woman's anxiety, depression, hope, self-efficacy and demographic data (age, place of residence). It turned out that basic hope is considered a fundamental component of women's worldview. Women report more loneliness than men, regardless of age. In addition, women live longer than men and are more likely to be affected by widowhood or take on the role of their spouse's guardian. At the same time, the prevalence of depression in women is higher than in men.
The study involved 207 women hospitalized during the pandemic.
The design of the study contained modern techniques that were chosen taking into account the risk of a COVID-19 pandemic, which is also very relevant today.
The central conclusion made by the authors of the article demonstrated various mediating roles of stress and understanding the relationship between the risk of infection with COVID-19, as well as personal resources with cognitive and affective aspects of subjective well-being. All these results show that hope and self-confidence, self-efficacy, allows not only to accelerate the recovery process from COVID-19, but also reduces anxiety and stress. The research has a scientific novelty, is interesting and is promising for the construction of new scientific hypotheses.

Author Response

Thank you for your valuable comments. With best regards!

Round 2

Reviewer 3 Report

Exclusion criteria have to be a subset of the inclusion criteria. Now the exclusion criteria are outside the inclusion criteria.

e.g. If inclusion criteria are between 18 to 83 years, the exclusion criteria cannot be outside that age range.

Author Response

Thank you for your valuable comment! We have clarified the inclusion and exclusion age criteria (line 130-133).
